# Anaphylaxis in Children and Adolescents: Greek Data Analysis from the European Anaphylaxis Registry (NORA)

**DOI:** 10.3390/jpm12101614

**Published:** 2022-09-30

**Authors:** Nikolaos Pyrpyris, Maria Kritikou, Xenofon Aggelidis, Ioanna Manolaraki, Michael Makris, Nikolaos G. Papadopoulos, Paraskevi Xepapadaki

**Affiliations:** 1Allergy and Clinical Immunology Unit, 2nd Pediatric Clinic, National and Kapodistrian University of Athens, 115 27 Athens, Greece; 2Allergy Unit, 2nd Department of Dermatology and Venereology, Attikon University Hospital, National and Kapodistrian University of Athens, 124 62 Athens, Greece

**Keywords:** anaphylaxis, food allergy, anaphylaxis management

## Abstract

The European Anaphylaxis Registry (NORA) was designed in order to create a comprehensive Anaphylaxis Registry regarding anaphylactic reactions characteristics and management in several European countries, including Greece. This study aims to analyze Greek data obtained in the NORA framework and describe anaphylaxis’ characteristics in this pediatric cohort. An online questionnaire was filled out by the Allergy Unit of the 2nd Pediatric Department of NKUA, regarding reported anaphylaxis characteristics. A total of 284 pediatric patients were analyzed. Patients were predominantly male. A previous, mild reaction (59.5%) to the same allergen was documented in 44.4% of patients. The most common system involved was skin and/or mucosa, followed by the respiratory system. Food was the most common eliciting factor (82.4%). First line treatment was most administered solely by a healthcare professional, followed by a lay person (38.7%). Auto injected adrenaline use by lay persons was third in frequency (29.2%). Most patients received post-reaction counseling and appropriate drug prescription. This study provides insight in anaphylactic cases’ characteristics and management in a Greek pediatric cohort. A low rate of adrenaline autoinjector administration was noted in lay people; however, drug prescription and counseling following the reaction could help increase correct acute anaphylaxis management in the future.

## 1. Introduction

Anaphylaxis is the most severe, potentially life-threatening systemic hypersensitivity reaction, with an estimated mortality rate lower than one per million people per year in developed countries [1]. In general, the prevalence of anaphylaxis ranges between 0.3–5.1%, with an incidence of 50–112 cases per 100,000 persons per year [2]. Regarding children, a recent systematic review by Wang et al. [3] reported that anaphylaxis incidence rates widely vary across the globe, ranging from 1 to 761 cases per 100,000 person-years. Despite concerns that anaphylaxis is underdiagnosed, it is anticipated that the introduction of the “anaphylactic and allergic reactions” indication in the International Classification of Diseases’ 11th edition (ICD-11) will contribute to the proper estimation of anaphylactic cases [1]. Pathophysiologically, anaphylaxis is mainly considered as an IgE mediated hypersensitivity reaction to certain allergenic proteins [4], although other less frequent types of anaphylactic reactions, the non-immunological, i.e., direct stimulation of mast cells and histamine secretion induced by some drugs [5], and idiopathic anaphylactic reactions have been described [6]. The most reported eliciting factors for children are food allergens, while for adults they are hymenoptera venoms and drugs [7]. According to the recently published study of the European Anaphylaxis Registry, the most common food allergens are hen’s egg, cow milk, and nuts [8]. During an anaphylactic reaction, symptoms from any organ may be present, however, the most implicated systems include the skin or/and mucosa, respiratory, cardio-circulatory, and gastrointestinal system [4]. Of note, fatal anaphylactic reactions can be associated with the absence of skin/cutaneous symptoms [9]. The first-line treatment according to the most recent guidelines from the European Academy of Allergy and Clinical Immunology (EAACI) [10] is intramuscular epinephrine, while second-line treatments include oxygen supplementation, corticosteroids, antihistamines, and inhaled β2 agonists.

Even though there are published data on the characteristics of anaphylactic reaction in different settings, respective data in the pediatric population in Greece are lacking. The purpose of this analysis is to describe the characteristics of anaphylactic reactions in a pediatric cohort, and to assess the compliance in anaphylaxis management in regards with the most recent Anaphylaxis Guidelines by EAACI. Children were examined in specialized allergy centers in Athens, Greece, and respective data were registered in the European Anaphylaxis Registry (NORA), a European registry including patients with a history compatible of an anaphylactic reaction, over a 14-year time period.

## 2. Materials and Methods

Within the European Anaphylaxis Registry, data from children with a reaction compatible with an anaphylaxis diagnosis were filled in online by the allergist who assessed, either at the time of the reaction or at a scheduled follow-up visit at the Allergy Unit, 2nd Pediatric Department, University of Athens, within 12 months during 2008–2021. In Greece, there are two specialized centers for children with allergy-associated diseases. However, the Allergy Unit of the 2nd Department of Pediatrics is the largest referral center for children with allergies, with more than 5000 visits annually. The raw data were recorded in the registry’s database, in compliance with the General Data Protection Regulation rules (GDPR). An informed consent was signed by all patients’ parents. Data were filled in the database the English language provided that the sex and year of reaction were known and the age at the time of the reaction was under 18 years. Variables were assorted in six broad categories, each corresponding to a different clinical aspect of the reaction and to the management of the anaphylactic episode. In respect to personnel providing the treatment during the anaphylactic reaction, two groups were created: the healthcare professionals, including doctors, nurses, and paramedics, and lay people, such as family members, who have no medical training. First line treatment was defined as emergency, first aid treatment until stabilization of the patient was achieved or advanced resuscitation was commenced. The database was updated yearly, for improvement of the domains, following experts’ consideration. For the present analysis, data were retrieved retrospectively by the team members, who also had access to the medical records of the patients. All data obtained were cleaned, using descriptive statistics with Fisher’s exact test, SPSS software (v. 28.0.1.0). Three age groups were created: neonates and infants including children <2 years of age: infancy *n* = 99, children aged from 2–12, *n =* 161 (childhood), and children ≥12 years (adolescents, *n =* 24), based on FDA’s Pediatric Exclusivity Study Age Groups. A *p*-value under 0.05 (*p* < 0.05) was considered as statistically significant.

## 3. Results

The cohort included 284 children, mean aged 5.5 (SD 0.26) years, 161 males (63.7%) with a history compatible of an anaphylactic reaction. In 28.5% of cases, the reaction occurred at home, with second most frequent site being a medical practice office or in a hospital. Previous immediate reaction to the same allergen was reported in 44.4% (*n =* 126) of the cases, with 45.7% reporting a reaction ≥3 times.

In respect to severity, previous reactions in majority were characterized as mild (*n =* 75, 59.5%), while in 21 cases (16.7%) they were considered as severe (according to Ring and Messner’s anaphylaxis severity classification [11]). However, in all age groups, previous reactions were most documented as less severe (infancy: *n =* 28, 22.2%, childhood: *n =* 43, 34.1%, adolescence: *n =* 4, 3.2%, *p* = 0.053).

In regard to the childhood group, the presence of a previous anaphylactic reaction and a diagnosis of the implicated factor were significantly more frequently (*n =* 120, 42,3%) than in the other age groups (infancy: *n =* 80, 28,2%; adolescence *n =* 13, 4,6%, *p* < 0.05). An already diagnosed allergy on the eliciting factor was statistically significant more frequently reported in the childhood group (*n =* 80, 28,2%), compared to the other two age groups (infancy: 31, 10.9%; adolescence:5, 1.8%, *p* = 0.007)

Most of the patients, *n =* 136, (63.6%) were admitted in the hospital, while 20 patients (9.3%) needed ICU treatment.

### 3.1. Symptoms/Organs Involved

Skin/mucosa was the most involved system in 245 patients (86.3%), with urticaria, angioedema, and erythema or flush being most commonly reported (*n =* 152, 62.0%; *n =* 94, 38.4%; *n =* 66, 26.9%, respectively). Symptoms from the respiratory system were reported in 222 cases (78.2%), while gastrointestinal symptoms and manifestations from the cardiovascular system were present in 48.2% (*n =* 137) and 15.1% (*n =* 43), respectively. Only one patient (0.5%) developed respiratory arrest. Data on the prevalence of symptoms during the anaphylactic reaction in the predefined age groups are provided in Table 1. Children aged 2–12 years presented significantly higher rates of cough, nausea, vomiting, and abdominal pain compared to the other age groups.

### 3.2. Implicated Allergens

In 213 children (75.0%), the eliciting factor has been identified, more so in infants, while in 59 (20.8%) there was a reasonable suspicion, deriving from the allergist’s evaluation. The most common eliciting factor was food allergens (*n =* 216, 82.4%), while insect venom (*n =* 16, 6.1%) and drugs (*n =* 9, 2.7%) were reported to a lower extent. Regarding foods, the most frequently identified categories were animal products (*n =* 113, 52.3%) and tree-nuts (*n =* 50, 23.2%), followed by cereals (*n =* 27, 12.5%) and peanuts (*n =* 22, 10.2%). In the insect venom category, bee venom (*n =* 8, 50.0%) was the most common, followed by hornet (*n =* 4, 25.0%). Regarding drugs, antibiotics (*n =* 4, 44.4%) and analgesics (*n =* 2, 22.2%) were most commonly responsible for an anaphylactic episode. Frequencies for the other allergens implicated in the anaphylactic reactions and in respect to the age groups are provided in Table 2.

### 3.3. Concomitant Diseases

Concomitant allergy-associated diseases were reported in 152 patients (53.5%) at the time of the reaction. Atopic dermatitis was the most frequent (*n =* 100, 65.8%), followed by food allergy (*n =* 38, 25.0%), asthma (*n =* 34, 22.4%) rhinitis (*n =* 32, 21,1%), while any kind of infections were present in 11 patients (7.2%). Urticaria, diabetes, cardiovascular diseases, malignancies, mastocytosis, and thyroid were not reported in the cohort.

### 3.4. Management of Anaphylaxis

The majority of the patients (*n =* 248, 87.3%) received first line treatment, either by a healthcare professional in 55.7% of cases (*n =* 138), who was more commonly an allergy specialist (*n =* 68, 49.3%) or by a lay, not qualified person, in 96 cases (38.7%). In most cases where a lay person administered any kind of treatment, it was a family member (*n =* 93, 96.9%). Treatment initially administered by a lay person and subsequently by a healthcare professional was documented in 13 cases (5.2%). Intramuscular (i.m.) adrenaline was mainly provided by professionals in 86 patients (62.3%), while only in 28 cases (29.2%) lay persons offered adrenaline. Other medications used by professionals and lay persons were oral antihistamines (47.8% vs. 79.2%), corticosteroids (49.3% vs. 41.7%), β2 agonists (23.2% vs. 10.4%), while the respective percentages for drug administration by professionals were oral antihistamines: 47.8%, oral corticosteroids: 34.8%, inhaled β2 agonists: 18.1%, and intravenous corticosteroids (iv): 14.5%. A second dose of adrenaline was administered in 4.5% of the reactors. Reasons for not using adrenaline in the lay persons were lack of previous prescription (64.5%) and/or, even if available, not used (26.5%),

A total of 87.4% of the patients received counselling regarding management of a potential new anaphylactic reaction, while adrenaline was prescribed according to recorded data in 75.2% of the children. Other commonly prescribed drugs were corticosteroids and antihistamines. Furthermore, most patients and their parents or tutors received emergency management training (85.9%), in order to be prepared and best know how to react in a future anaphylactic reaction.

## 4. Discussion

In this first descriptive analysis of anaphylaxis in children in Greece, we found that anaphylactic episodes are most common in males, presenting more frequently with skin and respiratory symptoms, while food allergens are the most implicated allergens in these reactions. Emergency management of the anaphylactic reaction was provided in 55.7% of the cases by a healthcare professional or in case of a lay person by a family member (96.9%). It is important to mention that there is a low percentage (29.2%) of adrenaline autoinjector by lay persons. However, we noticed a sufficient percentage of adrenaline i.m. use by healthcare professionals (62.3%). After the reaction, most of our patients received proper counselling and drug prescription.

A male predominance was identified in our cohort, which is also a frequent finding in previous studies and case series [12,13,14,15]. In our population, symptoms mainly involved the skin and respiratory system, consistent with previous data [1,8,16]. Approximately 10% of the children developed cardiovascular symptoms, which represent the most serious aspect of the anaphylactic reaction [17].

Regarding eliciting factors, we also found that the most frequent eliciting factor in all age groups is food, in accordance with published data [3,12,13,18,19,20], with animal products being more frequently recognized as in studies from Europe, Turkey, and Latin America [8,12,15,21]. Nevertheless, in a Korean study [13], tree nuts were as commonly identified in children <12 years, as milk in our population, potentially reflecting differences in the dietary habits on this population, while shrimp in adolescents [21,22,23]. Tree nuts are common food implicated allergens in anaphylactic reactions in our population, as in other Pediatric groups. Tree nuts were identified in 23.2% of the cases as was previously shown [3,19,23,24]. However, in our cohort, it was not the main elicitor for food anaphylaxis, despite other studies having tree nuts as the main eliciting factor [18,19]. In all age groups, cashew was the most common elicitor within tree nuts, in agreement with recent data deriving from NORA, and supporting that cashew and hazelnut have almost identical incidence rates [8]. Insect venoms are the second most frequent eliciting factor in our cohort (>2 years) consistent with previous NORA analyses [8,19], although the incidence was higher in our study [12,15,21,24]. Drug-induced anaphylaxis is the third cause of anaphylaxis as in other European studies [8,19,24], confirming the lower incidence of drug-triggered episodes in children, although not in all studies [12,21,23].

It is well established that epinephrine should be administered in anaphylactic episodes [10]. In our cohort, a total of 29.2% received autoinjected adrenaline from a lay person and 62.3% received adrenaline i.m. from a healthcare professional. This analysis shows that adrenaline is underused, especially by lay people. In healthcare professionals, the rate of adrenaline use is sufficient, as 62.3% of the cases treated by a healthcare professional received the appropriate treatment. Low rates of autoinjected adrenaline use are also mentioned by other studies [12,19,21,25,26], even when it is available for use [27]. The main reasons for not using an adrenaline autoinjector in our cohort were lack of previous prescription and decision to not use, regardless of its availability. In accordance, Curtis et al. [28] reported that adrenaline autoinjector was available for use in less than 60% of cases in a tertiary academic medical center, comparable with the 64.5% of cases with no previous prescription in this population. After the anaphylactic episode, however, approximately three out of four patients received an adrenaline autoinjector prescription, in accordance with EAACI Anaphylaxis guidelines.

Follow-up guidance after an anaphylactic episode is essential in order to investigate triggers, prevent a subsequent episode in the future, and educate the patient and their family in early recognition of anaphylaxis and its management, therefore as it is proposed by international guidelines and consensus on anaphylaxis [10,29]. The majority (87.4%) of our patients received an appointment with a specialist in order to investigate their anaphylactic reaction and discuss about the reaction and its management in future acute settings.

Our study has certain limitations. Data are derived from a single center, and the limited number of included children that do not allow generalization to the general population. Nevertheless, our center is the largest referral center for children with allergies in Greece, and a member of GALEN’s centers of excellence.

## 5. Conclusions

In this first analysis of the Greek data from the European Anaphylaxis Registry (NORA), we identified the characteristics of anaphylactic reactions in this pediatric cohort. Males experience anaphylactic reactions more frequently, while the main triggers are food, followed by insect venom and drugs. The rates of adrenaline use, especially by lay people, common, or non-qualified persons remains low, although increased adrenaline prescriptions and follow up appointments with allergy specialists were counseled indicated. There is a continuous need for better education of health personnel and lay, common people for the prompt management of anaphylactic reactions in all age groups.

## Figures and Tables

**Table 1 jpm-12-01614-t001:** Anaphylactic reactions’ symptoms, in regard to age.

Symptoms	Under 2	2–12	Over 12	*p*-Value
**Skin/mucosa (*n =* 245/284)**	90 (31.7%)	133 (46.8%)	22 (7.7%)	0.134
**Urticaria**	60 (24.5%)	78 (31.8%)	14 (5.7%)	0.474
**Angioedema/Laryngeal edema**	30 (12.3%)	59 (24.1%)	5 (2.0%)	0.072
**Gastrointestinal tract (*n* = 137/284)**	43 (15.1%)	83 (29.2%)	11 (3.9%)	0.098
**Abdominal pain/cramps**	8 (5.8%)	37 (27.0%)	7 (5.1%)	**0.002**
**Vomiting**	36 (26.3%)	54 (39.4%)	6 (4.4%)	**0.035**
**Nausea**	0 (0.0%)	12 (8.8%)	3 (1.2%)	**0.003**
**Respiratory system (*n =* 222/284)**	74 (26.1%)	128 (45.1%)	20 (18.8%)	0.860
**Dyspnea/SOB**	19 (8.6%)	43 (19.4%)	7 (3.2%)	0.448
**Cough**	38 (17.1%)	71 (32.0%)	5 (2.3%)	**0.043**
**Hoarseness**	15 (6.8%)	33 (14.9%)	6 (2.7%)	0.560
**Rhinitis**	33 (14.9%)	47 (21.2%)	8 (3.6%)	0.544
**Wheezing**	14 (6.3%)	28 (12.6%)	2 (0.9%)	0.451
**Cardiovascular system (*n =* 43/284)**	16 (5.6%)	23 (8.1%)	4 (1.4%)	0.132
**Hypotension**	4 (9.3%)	10 (23.3%)	1 (2.3%)	0.448
**Reduction of alertness**	8 (18.6%)	6 (14.0%)	1 (2.3%)	0.277
**Dizziness**	5 (11.6%)	7 (16.3%)	2 (4.7%)	0.736
**Loss of consciousness**	2 (4.7%)	3 (7.0%)	0 (0.0%)	0.747
**Chest pain/angina**	0 (0.0%)	0 (0.0%)	1 (2.3%)	0.093

Bold: significant *p* values.

**Table 2 jpm-12-01614-t002:** Implicated allergens.

Elicitor Factor	Under 2	2–12	Over 12	*p*-Value
**Identified elicitor factor**	80 (28.2%)	120 (42.3%)	13 (4.6%)	0.006
**Food (*n =* 216)**	89 (34.0%)	117 (44.7%)	10 (3.8%)	0.079
**Animal Products (*n =* 113)**	60 (27.7%)	45 (20.9%)	4 (1.9%)	-
**Cow’s milk**	41 (36.3%)	18 (15.9%)	0 (0.0%)	-
**Hen’s egg**	20 (17.7%)	7 (6.2%)	0 (0.0%)	-
**Codfish**	0 (0.0%)	6 (5.3%)	1 (0.9%)	-
**Shrimp**	0 (0.0%)	0 (0.0%)	2 (1.8%)	-
**Treenuts (*n =* 50)**	8 (16.0%)	39 (78.0%)	3 (6.0%)	-
**Cashew**	5 (10.0%)	15 (30.0%)	1 (2.0%)	-
**Pistachio**	1 (2.0%)	10 (20.0%)	0 (0.0%)	-
**Hazelnut**	1 (2.0%)	5 (10.0%)	1 (2.0%)	-
**Cereals (*n =* 27)**	9 (4.2%)	17 (7.9%)	1 (0.5%)	-
**Wheat flour**	8 (29.6%)	16 (59.3%)	0 (0.0%)	-
**Rye flour**	0 (0.0%)	1 (3.7%)	0 (0.0%)	-
**Barley**	0 (0.0%)	0 (0.0%)	1 (3.7%)	-
**Amaranth**	0 (0.0%)	1 (3.7%)	0 (0.0%)	-
**Peanuts/Legumes (*n =* 22)**	7 (3.2%)	13 (6.0%)	2 (0.9%)	-
**Peanut**	2 (9.1%)	7 (31.8%)	1 (4.5%)	-
**Lentil**	3 (13.6%)	1 (4.5%)	0 (0.0%)	-
**Soy**	0 (0.0%)	2 (9.1%)	1 (4.5%)	-
**Drugs (*n =* 9)**	1 (11.1%)	6 (66.7%)	2 (22.2%)	-
**Antibiotics**	0 (0.0%)	2 (22.2%)	2 (22.2%)	-
**Analgesics**	0 (0.0%)	2 (22.2%)	0 (0.0%)	-
**Insect venom (*n =* 16)**	0 (0.0%)	14 (87.5%)	2 (12.5%)	-
**Bee**	0 (0.0%)	8 (50.0%)	0 (0.0%)	-
**Yellow jacket**	0 (0.0%)	2 (12.5%)	1 (6.3%)	-
**Hornet**	0 (0.0%)	3 (18.8%)	1 (6.3%)	-
**Bumble bee**	0 (0.0%)	1 (6.3%)	0 (0.0%)	-

## Data Availability

Data available upon request.

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
