# Peer review of "Anaphylaxis in Children and Adolescents: Greek Data Analysis from the European Anaphylaxis Registry (NORA)"

_jpm, 2022, doi:10.3390/jpm12101614_

Round 1
Reviewer 1 Report
The lack of epidemiological data for anaphylaxis, especially for the pediatric age group, is of substantial importance to medical specialists, patients, and policymakers. Any piece of information is important for enlightening the related authorities and subjects. The manuscript provides substantial and up-to-date information about pediatric anaphylaxis in Greece, which, due to the standardized platform (NORA), is compatible with the data from the other European countries, which grants it additional importance.
The manuscript reviewed provides comprehensive and, at the same time, concise information about anaphylaxis in pediatric patients in Greece. The data collected covers a significant period of 14 years. One of the shortcomings of this type of collecting data is that it relies on self-reported cases or referred patients, which in fact excludes lethal cases. This is reflected in this manuscript – it is strange that there is not even one lethal case of such a severe condition for 14 years period. Since the database includes only patients referred to the NORA centers, this data cannot provide epidemiological data for the general population. However, it allows a detailed analysis of the causes, environmental setting, and treatment of anaphylactic cases. The manuscript provides a thorough and detailed analysis of a data set.
I consider the statistics applied appropriate for this type of study.
The conclusions derived from the data analysis are similar to other articles based on the NORA data, which resembles the generalized type of issues related to anaphylaxis diagnosis and treatment worldwide. They also can help to elaborate more personalized care for patients with anaphylaxis from different age groups and clinical profiles. The finding, that epinephrine i.m. use is underrated both by lay helpers and medical units may be an additional argument for the need for more education and search for new forms of its application.
The manuscript can be published in MDPI with minor technical corrections (e.g. Row 46: system [4], Row 174: [8,11,14,21], Row 164 - "i.m."
Reviewer 2 Report
Line 33- “Considering children ....ranging from 1 to 761 cases per 100 000 person‐years”.
Doubt -- the incidence in this sentence refers to children ?
If yes, it should be changed from "person" to children
Line 46 -cardio-circulatory and gastrointestinal system4
the 4 is a reference or just a mistake?
Line 47 - Of note, fatal anaphylactic reactions are can be associated with the absence of skin/cutaneous symptoms
S-fatal anaphylactic can be ...(withraw "are")
Line 53--- respective data in in the pediatric population
2 times in
Line 56- “Children were examined in specialized Allergy centers in Athens”
S-It was interesting to mention how many specialized centers you have in Athens
Line 90-91- (according to Ring anaphylaxis severity classification).
S-here the reference should be included and cited as Ring and Messner in the text
Line 94-96-”In regards to the childhood group, the presence of a previous anaphylactic reaction and a diagnosis of the implicated factor were significantly more frequently (n= 120, 42,3%) than in the other age groups (infancy: n=80, 28,2%; adolescence n=13, 4,6%, p<0.05)”.
S-The pragraph as it was written -not clear. Accept the changes marked as yellow to clarify
Line 101- here (n=152, 62,0%, n=94, 10438,4%, n=66, 26,9% respectively)
S- to separate de groups you should use ; and use . Instead of , eg- 26,9% must be 26.9
Line 110- “higher rates of cough, nausea”,
Table 1- only the p values that are statistically significant should be in bold
Line 116 - In 213 children (75%), the eliciting factor has been identified., more so in infants
S- modifie the sentence as sugested
Line 121-122-”In the insect venom category, bee venom (n=8, 50%) was the most common, followed by hornet (n=4, 25%)”.
S- Have you looked for mastocytosis in this group?
Line 131 - were reported in 152 patients
Line 133 - while any kind of infections were present in 11 patients (7.2%)
Line 135- "mastocytosis and thyroid were not reported in the cohort"
S- have you specifically looked for mastocitosis in the ones with anaphylaxis ?? specially those related to venoms ? this is an important point as anphylaxis is rare with venoms under 14-16 y of age
line 137- treatment),
Line 138- or who was more commonly an allergy specialist (n=72, 47.7%)
Line 137-139- The majority of the patients (n= 248 , 87.3%) received first line treatment either by a healthcare professional in 48,6% of cases (n=138), who was more commonly an ,allergy specialist (n=72, 47.7%)or and by a lay person not qualified person , in 96 cases (33.8%)
NOTE THAT - the percentages in this paragraph are NOT correct - if you take the total of treated as 248 the first, by healthcare pro is ..55.6 instead of 48.6.. and the second is 52.1% and 96 does not correspond to 33,8 of the 248 first line treated . The % is 38.7
In fact, all the results on the section of "management of anaphylaxis" must be revised by the authors
ALSO note that throughout the paper correct these numbers
you must use always the same signaling. When reading an English paper you don´t see 45,6% but always a dot so is 45.6%
Use this . whenever you have a ,
Line 154- "most patients "
S- you should write "most patients and their parents or tutors".…
we can not teach children of less than 12 and even after we can teach but aqlways the parents , tutors and give the instructions for them instruct close friends and relatives
Line 163-164- "It is important to mention that there is low percentage (31,3%) of adrenaline autoinjector by lay persons, but sufficient adrenaline im use by healthcare professionals 164
(68,1%)."
- This paragraph is not clear and i don´t quite understand what you have in mind to say . So, i suggest that an English native speaker helps in the English corrections that the paper also needs
Line 174 -" Latin America8,11,14,21."
-Latin America (8,11,14,21).
Discussion section -- all the percentages on the discussion area must be revised in accordance on my sugestion on the section of results and management and only after you can correct the discussion área
Line 170 - “children developed symptoms from the cardiovascular system”
S- children developed cardiovascular symptoms
Line 172- “we confirm that the most frequent eliciting factor “
S- we also found that the most frequent elicitin factor…….in accordance with published data
Line 174 -" Latin America8,11,14,21."
-Latin America (8,11,14,21).
Line 174-176 -”Nevertheless, in a Korean study [12] , tree nuts were as commonly identified in children <12 years, as milk in our population, potentially reflecting differences in the dietary habits on this population, 176 and shrimp in adolescents [21,22,23] “
S- accept the sugestion . The results of your study already described elsewhere
Line 179- “of the cases as was previously shown [3,19,23,24 “
Line 185- “although incidence rated are higher”
S- although the incidence was higher in our study
Line 189- “is the third cause of anaphylaxis as in other European studies “
S- as in other ….
Line 189 -It is well established that epinephrine should be administered in anaphylactic episodes [10]. In our cohort, a total of 31,3% received autoinjected adrenaline from a lay person and 68,1% received adrenaline im from a healthcare professional. This analysis show that 191 adrenaline is underused, especially by lay people
Note revise this %s and correct them in this section
Line 205- “therefore as it is proposed by international guidelines and consensus on anaphylaxis “
S- as it is proposed…..
Line - “about the reaction and its management in future acute settings “
Line 217-220-”The rates of adrenaline use, especially by lay people common or non qualified persons, remains low, although increased adrenaline prescriptions and follow up appointments with allergy-specialists were counseled indicated . There is a continuous need for better education of health personnel and lay people common people for the prompt management of anaphylactic reactions in all age groups. “
S- correct as above
Line 222-”Supplementary Materials “-
S - Supplementary informations
